# Beyond Atoms: Enhancing Molecular Pretrained Representations with 3D Space Modeling

**Shuqi Lu** [1]  **Xiaohong Ji** [1]  **Bohang Zhang** [2]  **Lin Yao** [1]  **Siyuan Liu** [1]  **Zhifeng Gao** [1]  **Linfeng Zhang** [1]  **Guolin Ke** [1]

## Abstract

Molecular pretrained representations (MPR) has emerged as a powerful approach for addressing the challenge of limited supervised data in applications such as drug discovery and material design. While early MPR methods relied on 1D sequences and 2D graphs, recent advancements have incorporated 3D conformational information to capture rich atomic interactions. However, these prior models treat molecules merely as *discrete* atom sets, overlooking the space *surrounding* them. We argue from a physical perspective that only modeling these discrete points is insufficient. We first present a simple yet insightful observation: naively adding randomly sampled virtual points beyond atoms can surprisingly enhance MPR performance. In light of this, we propose a principled framework that incorporates the entire 3D space spanned by molecules. We implement the framework via a novel Transformer-based architecture, dubbed SpaceFormer, with three key components: (1) grid-based space discretization; (2) grid sampling/merging; and (3) efficient 3D positional encoding. Extensive experiments show that SpaceFormer significantly outperforms previous 3D MPR models across various downstream tasks with limited data, validating the benefit of leveraging the additional 3D space beyond atoms in MPR models.

## 1. Introduction

Molecular pretrained representation (MPR) has been a key area of research for its crucial role in utilizing limited supervised data, particularly in real-world applications such as drug design and material discovery (Gilmer et al., 2017; Rong et al., 2020). The high cost of acquiring labeled data from wet-lab experiments or simulations limits the performance of models trained on small datasets. MPR addresses this by leveraging unsupervised pretraining on large-scale unlabeled data to learn molecular representations, which can be fine-tuned on limited labeled data for improved performance. The evolution of this field has progressed from 1D sequences (Xu et al., 2017; Wang et al., 2019; Heller et al., 2015) and 2D graphs (Hu et al., 2019; Rong et al., 2020; Li et al., 2021; Wang et al., 2022b) to 3D conformations (Stärk et al., 2022; Zhou et al., 2023), incorporating increasingly rich physical information and achieving superior performance. In all these prior 3D MPR models, atoms play a central role. More specifically, these models take the types and 3D positions of *atoms* (or *atom tuples*) as inputs and focus on modeling atomic interactions within 3D space, typically using graph neural networks or transformers (Zhou et al., 2023; Feng et al., 2023; Wang et al., 2023; Cui et al., 2024; Yang et al., 2024).

While this atom-based MPR approach appears straightforward, we argue that it has an inherent limitation: *it does not model the surrounding space beyond atoms*. On the surface, one may feel that the "empty" surrounding space is inconsequential as the molecular structure can be fully determined given only atom positions. However, such an argument overlooks critical physical principles. In the realm of microscopic physics, the space beyond atoms is *not* truly empty; it is permeated by electrons, various electromagnetic fields, and quantum phenomena (Atkins & Friedman, 2011; Zee, 2010; Weinberg, 1995). Moreover, many computational simulation methods in physics require modeling the entire 3D space, not just the positions of atoms. For instance, electronic density distributions and potential fields, which is closely related to certain molecular properties, are functions of the entire 3D space (Atkins & Friedman, 2011; Parr et al., 1979; Szabo & Ostlund, 2012). This naturally raises the following question:

> *Can modeling the 3D space beyond atoms improve molecular pretrained representation?*

To test the hypothesis, we start with a simple experiment. Given the point set of a molecule, we introduce randomly sampled virtual points (VPs) within the molecule's sur-

---

*Equal contribution  [1]DP Technology, Beijing, China  [2]Peking University, Beijing, China.  Correspondence to: Guolin Ke <kegl@dp.tech>.

*Proceedings of the $42^{nd}$ International Conference on Machine Learning*, Vancouver, Canada. PMLR 267, 2025. Copyright 2025 by the author(s).

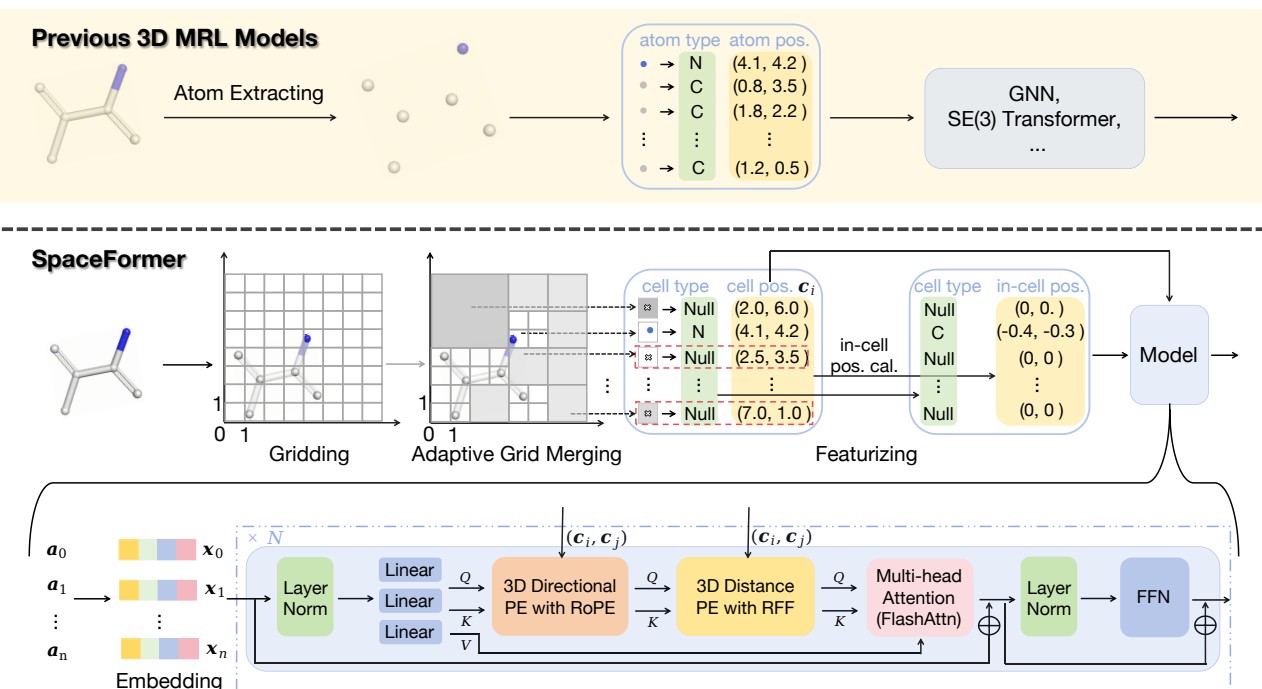

*Figure 1.* Overview of SpaceFormer. For clarity, we illustrate the model in 2D space. SpaceFormer begins by discretizing the 3D cuboid containing molecules into grid cells. Grid sampling/merging strategy is then applied to reduce the number of input cells for efficiency. Note that exact atomic positions are still encoded after discretization owing to various positional encoding (PE). To efficiently encode pairwise positional relationships, SpaceFormer utilizes 3D Directional PE with RoPE and 3D Distance PE with Random Fourier Features.

rounding space and feed the augmented point set to an existing 3D MPR model. Surprisingly, we observe a non-trivial performance gain after adding a small number of VPs, even though these VPs are inherently noisy and theoretically carry no meaningful information. Notably, the improvement remains consistent and gradually increases as the number of VPs grows, eventually reaching a plateau (see Fig. 2).

This finding is intriguing and further motivates us to develop a more principled approach that directly models the *entire* 3D space spanned by molecules. To this end, we propose a fundamentally different MPR framework: we treat a molecule as a 3D "image" by discretizing the 3D space into small grid cells. By using a sufficiently small cell size, we ensure that each grid cell contains at most one atom. The input feature for each cell can then be defined as the position of the cell combined with the type and position of the atom it contains (if any). This grid-based representation can subsequently be processed by any model capable of handling sets, such as Transformers.

**Efficient implementation.** One potential drawback of the above framework lies in its efficiency. Indeed, the number of grid cells grows cubically as the cell length decreases, which limits the scalability. To address this issue, we implement two intuitive strategies that significantly reduce the number of cells. The first one is *grid sampling*, where a random subset of empty grid cells is selected; The second

one is *adaptive grid merging*, where adjacent empty cells is recursively merged into large cells (see Fig. 1). Both methods work well in practice, reducing the number of cells by approximately an order of magnitude while still maintaining performance. Additionally, we design efficient versions of pairwise positional encoding based on RoPE (Su et al., 2024) and random Fourier features (Rahimi & Recht, 2007), which enjoy linear complexity relative to the number of cells. We term the resulting architecture **SpaceFormer**.

**Pretraining approach.** A key advantage of our proposed framework is that it can be integrated with a powerful pretraining approach known as Masked Auto-Encoder (MAE) (He et al., 2022). Specifically, during pretraining, we can mask a portion of grid cells for each molecule and let the model predict whether each masked cell contains atoms. If it is true, the model further predicts the atom type and its precise position within the cell. Remarkably, our analysis reveals a potential advantage of MAE pretraining compared with existing approaches such as denoising (Zaidi et al., 2022), showing that the model may learn additional knowledge via masked prediction.

We conduct extensive experiments to evaluate the effectiveness and efficiency of SpaceFormer. Across a total of 15 diverse downstream tasks, SpaceFormer achieves the best performance on 10 tasks and ranks within the top 2 on 14 tasks. Ablation studies further confirm that each component

plays a critical role in enhancing either the performance or efficiency of SpaceFormer. Overall, these results underscore the importance of modeling the 3D space beyond atom positions, and we believe our framework can serve as a new paradigm towards advancing the area of MPR.

## 2. Related Work

**Molecular Pretrained Representation.** Molecular pretrained representation has explored various modalities, giving rise to a diverse range of methods that leverage different molecular input formats. Some approaches rely on 1D sequences, such as SMILES (Wang et al., 2019; Xu et al., 2017), while others focus on 2D topologies, including methods like MolCLR (Wang et al., 2022b), MolGNet (Li et al., 2021), ContextPred (Hu et al., 2019), GROVER (Rong et al., 2020). Additionally, several works have enhanced 2D MPR models by integrating 3D information, as seen in approaches like GEM (Fang et al., 2022), 3D-Infomax (Stärk et al., 2022), MoleBLEND (Yu et al., 2024), GraphMVP (Liu et al., 2021), and Transformer-M (Luo et al., 2022).

Recently, starting with Noisy Nodes (Zaidi et al., 2022) and Uni-Mol (Zhou et al., 2023), pure 3D MPR models have demonstrated superior performance across various tasks. This has inspired a surge of subsequent works (Feng et al., 2023; Wang et al., 2023; Cui et al., 2024; Yang et al., 2024), which further explored the potential of 3D MPR. Besides, several other works also focus on 3D conformations, such as deep potential models (Schütt et al., 2017; Thomas et al., 2018; Gasteiger et al., 2020; 2021; Liu et al., 2022; Wang et al., 2022a; Jiao et al., 2023), protein folding (Jumper et al., 2021; Abramson et al., 2024), and 3D conformation generation (Shi et al., 2021; Zhu et al., 2022b; Xu et al., 2022; 2021). Yet, all of these works use 3D atomic positions as inputs without modeling the empty space beyond atoms. Earlier 3D grid-based methods like AtomNet (Wallach et al., 2015) and SchNet (Schütt et al., 2017) both use 3D convolutional neural networks, with SchNet focusing on atom positions. In contrast, SpaceFormer employs a transformer-based architecture to capture interactions across global grid cells, extending beyond just atoms positions. Separately, research on potential energy surfaces (Wang et al., 2024; Wallach et al., 2015) prioritizes long-range and higher-order atomic interactions for force field applications; SpaceFormer, however, is not optimized for such force-specific tasks, distinguishing its scope and objectives.

**The use of virtual points and auxiliary tokens.** Virtual points has been extensively used in the area of 3D point cloud learning to serve as intermediate representations. For example, Wu et al. (2023); Yin et al. (2021) converted 2D camera images into 3D virtual points, which are then fused with 3D LiDAR points to create a unified input representation. Similarly, Zhu et al. (2022a) introduced sparse virtual points to align and fuse features from 2D camera images and 3D LiDAR data, effectively addressing the resolution disparity between the sensors. Various studies (Song et al., 2023a;b; Mahmoud et al., 2023) have advanced this direction by utilizing virtual points as a bridge to align data from heterogeneous sensors. In this paper, virtual points are used in a quite different context, in the sense that it is a starting point to motivate our grid-based SpaceFormer architecture.

Broadly speaking, the idea of adding virtual points can be generalized to the use of auxiliary tokens, a trick that has been adopted across various domains beyond 3D point clouds, such as vision and language. In Vision Transformers, Darcet et al. (2023) introduced additional "register" tokens into input sequence patches to mitigate artifacts. Similarly, in language modeling, Pfau et al. (2024) demonstrated that incorporating dot tokens ("...") as chain-of-thought prompts surprisingly enhances the performance of large language models. Our approach aligns partially with this design principle; However, the underlying mechanism is fundamentally distinct: the empty grid cells in SpaceFormer are guided by physical principles along with meaningful 3D positional encodings, and it enjoys MAE pretraining. In Sec. 4.3, we will demonstrate that these aspects contribute to the improved performance of our model.

## 3. Method

### 3.1. Motivation: the effect of adding virtual points

As mentioned in the introduction, we posit that modeling the surrounding space beyond atoms can enhance 3D MPR due to its ability to capture critical physical principles such as electromagnetic fields and quantum phenomena. To illustrate this, we design the following motivating experiment. Given an input molecule represented as a set of atoms, we introduce virtual points (VPs) randomly sampled from the 3D space within the circumscribed cuboid of the atom set. This augmented point set is then fed into a state-of-the-art 3D MPR model, Uni-Mol (Zhou et al., 2023). We follow exactly the same pretraining pipeline as described in Zhou et al. (2023), which involves randomly masking atom types and adding Gaussian noise to their positions, while training the model to predict the correct atom types and the noises. Our goal is to investigate how the model's performance varies with the number of random virtual points, where the baseline corresponds to no virtual points.

The result is illustrated in Fig. 2. One can clearly see that the validation loss gets improved with the help of virtual points. Specifically, adding just 10 virtual points already yields a noticeable performance gain. This improvement remains consistent and gradually increases as the number of virtual points (VPs) grows, eventually reaching a plateau. Such a result is perhaps surprising as the additional virtual

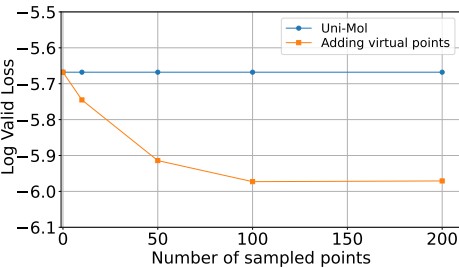

*Figure 2.* Log-scale valid loss of Uni-Mol when adding virtual points. The blue horizontal line represents the original Uni-Mol model, while the orange curve represents the Uni-Mol performance with respect to the number of virtual points.

points is purely random and can be interpreted as noise. In Sec. 4.3, we further demonstrate that the improved pretraining loss translates to improved performance on downstream tasks, and notably, the improvement is not attributed to the (slightly) increased computational costs.

While the above experiment confirms that simply adding virtual points can already enhance 3D MPR, it further raises the question of whether a more systematic approach might yield even better results. In the next section, we will explore this question by introducing a principled framework that directly models the entire 3D space.

### 3.2. Overall of Our framework

Our approach is inspired by the standard input format in computer vision, where inputs are uniformly sampled from the continuous 2D space and represented as discrete pixel grids. This can be naturally applied to our setting by discretizing 3D space into small, uniform grid cells. Compared to randomly sampling virtual points described in Sec. 3.1, the grid-based sampling is clearly better as it ensures uniform coverage of the 3D space. To choose a proper grid size (resolution), we use a fundamental physical principle: two distinct atoms must maintain a minimum separation distance due to repulsive forces. Consequently, as long as the cell length $c^l$ satisfies $c^l < \frac{\hat{d}}{\sqrt{3}}$, where $\hat{d}$ represents the minimum distance between any pair of atoms, the discretization will be accurate enough in the sense that each grid contains at most one atom. Note that $\hat{d}$ typically does not depend on specific tasks: for small organic molecules, $\hat{d}$ is approximately 0.96Å corresponding to the O-H bond length. We will fix $c^l = 0.49$Å throughout this paper.

**Data augmentation.** We employ two simple types of data augmentation. First, we randomly rotate the molecules prior to discretization to enhance the model's robustness to rotation. Second, after discretization, we randomly pad each face of the grid boundary by up to 2 grid cells, ensuring that the surrounding space of the atoms is further expanded to capture a broader spatial context.

**Input format.** Based on the discretization, the input to our framework is simply the set of grid cells, where each cell consists of two parts: its type and coordinates. Since each grid cell contains at most one atom, the cells can be categorized as either atom cells or non-atom cells. For each atom cell, its type and coordinates are defined as the atom type and its precise 3D coordinates. For each non-atom cell, we assign it with the special type NULL and define the coordinates to be its *cell center*. In the following, we will use $t_i$ and $c_i \in \mathbb{R}^3$ to denote the type and coordinates of the $i$-th cell, respectively. The coordinates will be further processed in the following embedding layer as well as relative positional encoding in Transformer layers.

**Embedding.** Given each cell represented as $(t_i, c_i)$, we process it with the following embedding layer. First, we compute the *inner* position of the atom (or cell center for non-atom cells) within the cell, defined as $\tilde{e}_i = c_i \mod c^l$, followed by discretization $e_i = \lfloor \frac{\tilde{e}_i}{c^m} \rfloor$, where $c^m$ is a hyperparameter setting to a very small value, i.e., $c^m = 0.01$Å . In this way, the resulting $e_i$ is an integer vector with elements in range $\left[0, \frac{c^l}{c^m}\right)$. Next, we convert all discrete input features $a_i := (t_i, e_i) \in \mathbb{N}^4$ into continuous feature representations by summing the corresponding embedding layers, i.e., $x_i = \sum_{t=1}^{4} \text{Embed}_t(a_{i,t})$, where $\text{Embed}_t(\cdot)$ is the parameterized embedding function that maps discrete inputs to continuous representations, and $x_i$ is the resulting input embedding for the $i$-th cell.

**Architecture.** Given the set of input features $\{x_i\}$, we can process it using any model capable of handling sets, in particular, the Transformer architecture (Vaswani, 2017). We call the resulting model SpaceFormer. However, a naive application of the SpaceFormer suffers from significant computational costs, as its complexity scales quadratically with the number of grid cells, which can be substantial due to the discretization process. For instance, in widely used organic molecule datasets like ZINC (Sterling & Irwin, 2015), the average number of cells is approximately 8,000. Such a high cost can be partially mitigated by using advanced attention implementation such as FlashAttention (Dao et al., 2022), which is adopted in this paper. Yet, it remains higher than that of previous atom-based methods.

In the subsequent subsections, we will propose concrete implementations that aim to significantly improves the efficiency of our proposed SpaceFormer.

### 3.3. Grid sampling and merging

To reduce computational costs, a straightforward direction is to reduce the number of non-atom cells. We propose two intuitive strategies to achieve this goal, as presented below.

**Grid sampling.** In this strategy, we just randomly select a portion of non-atom grid cells and feed the selected cells

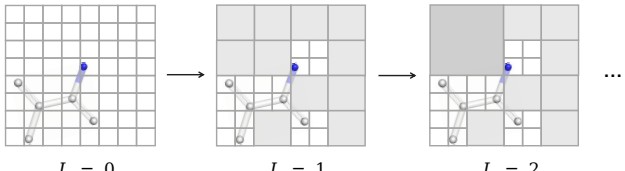

*Figure 3.* Adaptive grid merging. Dark cells represent merged cells, and $L$ denotes the number of merging steps.

along with all atom cells into the model. Empirically, we find that this strategy already works well when sampling only 10%-20% non-atom cells.

**Adaptive grid merging.** The potential limitation of the above sampling strategy is that it does not incorporate fundamental physical principles and involves tuning an additional hyper-parameter. Specifically, in quantum physics, regions close to atoms typically exhibit higher electron density, and the density within these regions can vary significantly with position. Conversely, regions far from all atoms generally have electron densities approaching zero. Consequently, computational simulations often employ fine-grained sampling near atoms to accurately capture these dynamic variations, while coarse-grained sampling is used in distant regions to optimize computational efficiency (Parr et al., 1979). Drawing inspiration from this approach, we introduce an adaptive grid merging strategy, as illustrated in Fig. 3. Specifically, starting from the full grid of cells, we group them by $2 \times 2 \times 2$ blocks and merge each block of eight adjacent cells into a "larger" cell if all of them are non-atom cells. This process can be executed recursively until convergence. After merging, the coordinate of each merged cell is still defined to be its cell center, while its type will further distinguish the cell size (or merging level). Empirically, we find that this strategy can reduce the number of grid cell by an order, while maintaining the performance without tuning any hyper-parameter.

### 3.4. Efficient 3D relative positional encoding

Positional encoding (PE) is critical in Transformer-based models since the original Transformer architecture lacks inductive bias for its input structure. While absolute positional information is typically injected into the input, relative PE is equally important as it captures pairwise relationships between grid cells. However, naively integrating relative PE is computationally inefficient due to its quadratic memory cost relative to the number of grid cells. This inefficiency becomes particularly impractical when combined with FlashAttention.

To address this issue, we propose an efficient positional encoding method tailored for continuous 3D coordinates. Given two points $A, B$ in 3D space, denote their coordinates as $c_A$ and $c_B$, respectively. The relative positional information can be defined as the vector $\overrightarrow{AB} := c_B - c_A$. In the

following, we propose two types of *linear-complexity* 3D positional encoding: the first directly encodes $\overrightarrow{AB}$, which captures the raw directional information; the second encodes the geometric distance $\|\overrightarrow{AB}\|_2$, which is a fundamental invariant under coordinate system transformations.

**3D directional PE with RoPE.** Recently, Rotary Positional Encoding (RoPE) (Su et al., 2024) has gained popularity due to its linear-complexity in encoding relative positions. Here, we extend RoPE to 3D continuous space to encode directional information $\overrightarrow{AB}$, capturing pairwise directional relationships across all three axes.

Recall that in RoPE, the Query and Key vector of each token is multiplied by a 2D position-aware rotational matrices before performing dot-product attention, which has the following form:

$$(\boldsymbol{R}(i)\boldsymbol{q}_i)^\top (\boldsymbol{R}(j)\boldsymbol{k}_j) = \boldsymbol{q}_i^\top (\boldsymbol{R}(i)^\top \boldsymbol{R}(j))\boldsymbol{k}_j$$
$$= \boldsymbol{q}_i^\top \boldsymbol{R}(j-i)\boldsymbol{k}_j,$$

where $\boldsymbol{q}_i$ is the Query vector of the $i$-th token, $\boldsymbol{k}_j$ is the Key vector of the $j$-th token, and $\boldsymbol{R}(i)$ is the $d \times d$ rotational matrix with parameters $\theta_1, \cdots, \theta_{d/2}$ defined below:

$$\boldsymbol{R}(i) = \begin{bmatrix} \cos i\theta_1 & -\sin i\theta_1 & & & \\ \sin i\theta_1 & \cos i\theta_1 & & & \\ & & \ddots & & \\ & & & \cos i\theta_{d/2} & -\sin i\theta_{d/2} \\ & & & \sin i\theta_{d/2} & \cos i\theta_{d/2} \end{bmatrix}.$$

To extend RoPE to 3D continuous space, we split each Query and Key vector into 3 blocks and apply a rotational matrix separately for each block. Formally,

$$\left( \begin{bmatrix} \boldsymbol{R}(c_{i,1}) & & \\ & \boldsymbol{R}(c_{i,2}) & \\ & & \boldsymbol{R}(c_{i,3}) \end{bmatrix} \begin{pmatrix} \boldsymbol{q}_{i,1} \\ \boldsymbol{q}_{i,2} \\ \boldsymbol{q}_{i,3} \end{pmatrix} \right)^\top \left( \begin{bmatrix} \boldsymbol{R}(c_{j,1}) & & \\ & \boldsymbol{R}(c_{j,2}) & \\ & & \boldsymbol{R}(c_{j,3}) \end{bmatrix} \begin{pmatrix} \boldsymbol{k}_{j,1} \\ \boldsymbol{k}_{j,2} \\ \boldsymbol{k}_{j,3} \end{pmatrix} \right)$$
$$= \sum_{s=1}^{3} \boldsymbol{q}_{i,s}^\top \boldsymbol{R}(c_{j,s} - c_{i,s})\boldsymbol{k}_{j,s}.$$

In this way, the directional information of each axis is encoded in the Transformer layer via $\boldsymbol{R}(c_{j,s} - c_{i,s})$.

**3D distance PE with RFF.** The geometric distance is a fundamental measure of pairwise interactions in quantum mechanics. To overcome the quadratic memory overhead associated with computing pairwise distances, we propose using random Fourier features (RFF) to approximate the Gaussian kernel (Rahimi & Recht, 2007), which has the

following form:

$$\exp\left(-\frac{\|c_i - c_j\|^2}{2\sigma^2}\right) \approx z(c_i)^\top z(c_j),$$

$$z(c_i) = \sqrt{\frac{2}{d}} \cos\left(\frac{c_i^\top \omega}{\sigma} + b\right),$$

$$\omega \in \mathbb{R}^{3 \times d} \sim \mathcal{N}(\mathbf{0}, \mathbf{I}), \quad b \in \mathbb{R}^d \sim \mathcal{U}([0, 2\pi)^d),$$

where $\sigma$ controls the bandwidth of the Gaussian kernel, each element of $\omega$ is i.i.d. sampled from standard normal distribution, and $b$ is sampled from uniform distribution over $[0, 2\pi)^d$.

The random Fourier features are then combined with the Query and Key after applying the RoPE:

$$\tilde{q}_i = [q_i^\top, z(c_i)^\top]^\top, \quad \tilde{k}_j = [k_j^\top, z(c_j))^\top]^\top.$$

Here, the concatenation ensures that the effect of RoPE and RFF are separated as $\tilde{q}_i^\top \tilde{k}_j = q_i^\top k_j + z(c_i)^\top z(c_j)$.

### 3.5. Pretraining

A key advantage of our proposed framework is that it admits a powerful and efficient pretraining approach known as Masked Auto-Encoder (MAE) (He et al., 2022). In MAE, we randomly mask a portion of (either atom or non-atom) grid cells and only feed the *remaining* cells into Space-Former to obtain the representation of these cells. Then, we use another decoder architecture that takes the representations along with all masked cells as inputs to predict their type and coordinates. The MAE approach is highly efficient because (1) the encoder operates only on unmasked cells, and (2) the decoder, which takes all cells as inputs, is significantly smaller than the encoder.

**Implementation details.** While the above pretraining framework is straightforward, several implementation details distinguish it from image pretraining, which we highlight below. First, we must handle the prediction of atom and non-atom cells differently. Note that the input coordinates of masked cells are defined by their center positions. Therefore, if a cell is predicted to be an atom cell, the model must additionally predict the offset relative to the cell center. Moreover, since the numbers of atom and non-atom cells are highly imbalanced, we assign different loss weights to the prediction of each cell type to ensure their overall contributions are balanced. Second, caution is required when combining MAE with adaptive grid merging, as masking merged grids does not make sense (since it can be trivially predicted). Instead, the masking process is applied prior to adaptive grid merging, and all masked cells are excluded thereafter in the encoder.

**Comparison with prior pretraining pipelines.** In the literature, most prior MPR approaches are based on *denoising*

(e.g., Zaidi et al., 2022; Zhou et al., 2023). We argue that our MAE-based pretraining could be more advantageous, because it involves a stronger training task that helps the model acquire additional knowledge during pretraining. Specifically, the prediction task in MAE can be decomposed into two components: the first involves predicting whether an atom exists within a certain range in 3D space, while the second refines the exact coordinates relative to the grid center. One can see that the second task is exactly a form of "denoising", whereas the first task is absent in denoising-based approaches. Moreover, unlike denoising, which *perturbs* certain atoms, we *remove* them entirely, making the pretraining task even harder. We hypothesis that these differences contribute to the improved performance, as will be demonstrated in the following section.

## 4. Experiments

To evaluate our proposed SpaceFormer, we first conduct unsupervised pretraining on large-scale unlabeled data, and then fine-tune on various tasks with limited labeled data. Moreover, we perform extensive ablation studies to assess the contribution of its key components. Finally, we compare the effectiveness and efficiency of SpaceFormer with atom-based MPR models that incorporate virtual points.

### 4.1. Settings

**Pretraining settings.** We use the same pretraining dataset as Zhou et al. (2023), which contains a total of 19 million molecules. Details of the pretraining settings are provided in Table 5 in the Appendix. For grid merging, we set the merging level to 3, which corresponds to convergence. This configuration results in a model with approximately 67.8M (encoder) parameters and requires about 50 hours of training using 8 NVIDIA A100 GPUs.

**Baseline models.** Our primary baseline is the Uni-Mol (Zhou et al., 2023), a recent 3D MPR model that achieved state-of-the-art performance across a wide range of molecular property prediction tasks. We use the same pretraining dataset as Uni-Mol to enable a fair comparison. We also include Mol-AE (Yang et al., 2024), which extends Uni-Mol with (atom-based) MAE pretraining. For a more comprehensive comparison, we further include 3D Infomax (Stärk et al., 2022), a GNN based representation incorporating 3D information, and two 2D graph-based MPR models: GROVER (Rong et al., 2020) and GEM (Fang et al., 2022).

**Downstream tasks.** We develop a new benchmark to comprehensively evaluate MPR models[1], particularly focusing

---

[1]Most prior works follow MoleculeNet (Wu et al., 2018) for downstream task evaluation. However, recent studies (Walters, 2023) identified several limitations of this dataset, including the presence of invalid structures, inconsistent chemical representa-

*Table 1.* Performance on molecular computational property and experimental properties prediction tasks. The best results are highlighted in **bold**, and the second-best results are underlined.

| Task | #Samples | GROVER | GEM | 3D Infomax | Uni-Mol | Mol-AE | SpaceFormer |
|---|---|---|---|---|---|---|---|
| HOMO (Hartree) ↓ | 20,000 | $0.0075 \pm$ 2.0e-4 | $0.0068 \pm$ 7.0e-5 | $0.0065 \pm$ 1.0e-5 | $0.0052 \pm$ 2.0e-5 | $\underline{0.0050} \pm$ 8.0e-5 | $\mathbf{0.0042} \pm$ 1.0e-5 |
| LUMO (Hartree) ↓ | 20,000 | $0.0086 \pm$ 8.0e-4 | $0.0080 \pm$ 2.0e-5 | $0.0070 \pm$ 1.0e-4 | $0.0060 \pm$ 6.0e-5 | $\underline{0.0057} \pm$ 4.7e-4 | $\mathbf{0.0040} \pm$ 2.0e-5 |
| GAP (Hartree) ↓ | 20,000 | $0.0109 \pm$ 1.4e-3 | $0.0107 \pm$ 1.9e-4 | $0.0095 \pm$ 1.0e-4 | $0.0081 \pm$ 4.0e-5 | $\underline{0.0080} \pm$ 8.0e-5 | $\mathbf{0.0064} \pm$ 1.2e-4 |
| E1-CC2 (eV) ↓ | 21,722 | $0.0101 \pm$ 9.7e-4 | $0.0090 \pm$ 1.3e-4 | $0.0089 \pm$ 2.0e-4 | $\underline{0.0067} \pm$ 40e-5 | $0.0070 \pm$ 6.0e-5 | $\mathbf{0.0058} \pm$ 8.0e-5 |
| E2-CC2 (eV) ↓ | 21,722 | $0.0129 \pm$ 4.6e-4 | $0.0102 \pm$ 2.3e-4 | $0.0091 \pm$ 3.0e-4 | $\underline{0.0080} \pm$ 4.0e-5 | $\underline{0.0080} \pm$ 40e-5 | $\mathbf{0.0074} \pm$ 8.4e-5 |
| f1-CC2 ↓ | 21,722 | $0.0219 \pm$ 3.5e-4 | $0.0170 \pm$ 4.3e-4 | $0.0172 \pm$ 4.0e-4 | $0.0143 \pm$ 2.0e-4 | $\mathbf{0.0140} \pm$ 4.0e-5 | $\underline{0.0142} \pm$ 3.7e-4 |
| f2-CC2 ↓ | 21,722 | $0.0401 \pm$ 1.2e-3 | $0.0352 \pm$ 5.4e-4 | $0.0364 \pm$ 9.0e-4 | $0.0309 \pm$ 9.4e-4 | $\underline{0.0307} \pm$ 1.3e-3 | $\mathbf{0.0294} \pm$ 7.1e-4 |
| Dipmom (Debye) ↓ | 8,678 | $0.0752 \pm$ 1.1e-3 | $0.0289 \pm$ 1.2e-3 | $0.0291 \pm$ 1.7e-3 | $\underline{0.0106} \pm$ 3.1e-4 | $0.0113 \pm$ 4.7e-4 | $\mathbf{0.0083} \pm$ 5.0e-4 |
| aIP (eV) ↓ | 8,678 | $0.1467 \pm$ 1.5e-2 | $0.0207 \pm$ 2.6e-4 | $0.0526 \pm$ 1.4e-4 | $\underline{0.0095} \pm$ 6.4e-4 | $0.0103 \pm$ 1.3e-4 | $\mathbf{0.0090} \pm$ 5.9e-4 |
| D3_disp_corr (eV) ↓ | 8,678 | $0.2516 \pm$ 5.3e-2 | $0.0077 \pm$ 6.6e-4 | $0.2285 \pm$ 7.5e-3 | $\mathbf{0.0047} \pm$ 5.6e-4 | $0.0077 \pm$ 1.3e-3 | $\underline{0.0053} \pm$ 1.2e-3 |
| HLM ↓ | 3,087 | $0.4313 \pm$ 3.2e-2 | $0.3319 \pm$ 3.3e-3 | $0.3333 \pm$ 9.0e-3 | $\underline{0.3100} \pm$ 8.2e-3 | $0.3127 \pm$ 1.5e-2 | $\mathbf{0.3098} \pm$ 7.4e-3 |
| MME ↓ | 2,642 | $0.4438 \pm$ 1.7e-2 | $0.3132 \pm$ 1.4e-2 | $0.3275 \pm$ 3.6e-3 | $\mathbf{0.2901} \pm$ 3.0e-3 | $0.2947 \pm$ 1.1e-2 | $\underline{0.2933} \pm$ 6.8e-3 |
| Solu ↓ | 2,173 | $0.2660 \pm$ 1.8e-2 | $\underline{0.2290} \pm$ 3.1e-3 | $0.4535 \pm$ 1.4e-2 | $0.3388 \pm$ 7.1e-3 | $0.2320 \pm$ 1.2e-2 | $\mathbf{0.2205} \pm$ 1.8e-3 |
| BBBP ↑ | 1,965 | $0.8672 \pm$ 2.6e-2 | $0.8863 \pm$ 2.5e-2 | $0.8276 \pm$ 2.7e-2 | $\mathbf{0.9066} \pm$ 2.3e-2 | $\underline{0.8995} \pm$ 4.6e-3 | $0.8605 \pm$ 2.2e-2 |
| BACE ↑ | 1,513 | $0.9053 \pm$ 2.8e-2 | $\mathbf{0.9851} \pm$ 2.8e-3 | $0.9612 \pm$ 7.4e-3 | $0.9523 \pm$ 2.0e-2 | $0.9605 \pm$ 1.3e-2 | $\underline{0.9657} \pm$ 3.1e-2 |

on the setting of limited data. Our benchmark comprises two categories of tasks: molecular computational properties and molecular experimental properties. For computational properties, we sample a 20K subset from the huge dataset of GDB-17 (Ramakrishnan et al., 2014; Ruddigkeit et al., 2012) and select the electronic properties HOMO, LUMO and GAP. Additionally, we use another 21k subset from the same dataset following (Ramakrishnan et al., 2015), selecting the energy properties E1-CC2, E2-CC2, f1-CC2 and f2-CC2. Furthermore, we incorporate the dataset (Wahab et al., 2022) of cata-condensed polybenzenoid hydrocarbons comprising from 1 to 11 rings which contain 8k samples and selected the mechanical and electronic properties within and between molecules, like Dipmom, aIP and D3 Dispersion Corrections (D3_disp_corr). For experimental properties, we select the BBBP and BACE datasets from MoleculeNet, ensuring that all duplicate and structurally invalid molecules were excluded. Additionally, we employ the HLM, MDR1-MDCK ER (MME), and Solubility (Solu) datasets from the Biogen ADME dataset (Fang et al., 2023). A detailed description of these tasks is provided in Table 7 in the Appendix. In all tasks, datasets were split into training, validation, and test sets in an 8:1:1 ratio. We applied the Out-of-Distribution splitting methods, where the sets are divided based on scaffold similarity. This eventually results in 15 tasks, allowing for a thorough evaluation. We ensure that the hyper-parameter search space is consistent across all tasks and baseline models (see Table 6 in the Appendix). For each set of hyper-parameters, we train each model 3 times using different random seeds and report the the average result and standard deviation. The checkpoint with the best validation loss is selected for all experiments.

tions, and data curation errors. Moreover, Sun et al. (2022) pointed out that MoleculeNet fails to adequately distinguish the performance of different molecular pretraining models. Instead, this paper proposes a new benchmark to address these problems.

### 4.2. Results

Table 1 present our experimental results. The results clearly demonstrate SpaceFormer's superior performance, which ranks the first in 10 out of 15 tasks and within the top two in 14 out of 15 tasks. In particular, SpaceFormer significantly outperforms all baselines in computational properties such as HOMO, LUMO, GAP, E1-CC2 and Dipmom tasks, where it surpasses the runner-up models by about 20%. This verifies the effectiveness of modeling 3D space beyond atom positions and justifies the design of SpaceFormer.

SpaceFormer has limited effectiveness on the BBBP task, one reason can be the complexity and variability of the measurement environments for experimental properties, which can be easily influenced by external factors, like temperature and blood-brain barrier biofilm status for BBBP. Such variability can affect the stability of measurement results and may not accurately. Additionally, as mentioned in the footnote of Section 4.1, the MoleculeNet dataset for the experimental task (including BBBP) has several limitations, including invalid structures and inconsistent chemical representations, which affect its ability to differentiate molecular pretraining models.

### 4.3. Ablation Studies

In this subsection, we conduct a series of experiments to evaluate the proposed components of SpaceFormer. We choose the HOMO, LUMO, E1-CC2, and E2-CC2 properties throughout all ablation experiments.

**The effect of grid sampling/merging.** As discussed in Sec. 3.3, we propose using either random sampling or adaptive grid merging for non-atom cells to reduce training costs. Here, we evaluate the efficiency and performance of both approaches by varying the number of input cells. For adaptive grid merging, we experiment with different merging levels $L$. A level of $L = 0$ indicates no merging (using

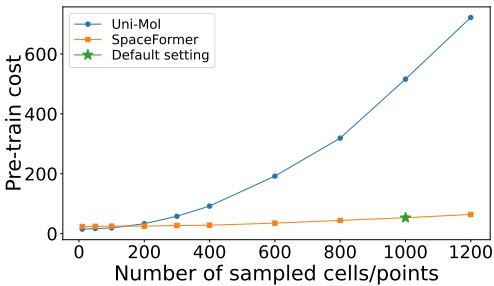

*Figure 4.* Efficiency comparison of SpaceFormer and Uni-Mol with different numbers of cells/points.

*Table 2.* Ablation studies on grid sample/merging strategy.

| No. | $L$ | Sample Ratio | HOMO↓ | LUMO↓ | E1-CC2↓ | E2-CC2↓ | Pretraining cost | avg. #cells |
|---|---|---|---|---|---|---|---|---|
| 1 | 0 | 1.0 | 0.0043 | 0.0041 | 0.0057 | 0.0073 | 656h | 8.4K |
| 2 | 1 | - | 0.0044 | 0.0041 | 0.0059 | 0.0073 | 122h | 2.2K |
| 3 | 2 | - | 0.0042 | 0.0043 | 0.0058 | 0.0075 | 56h | 1.1K |
| 4 | 3 | - | 0.0042 | 0.0040 | 0.0058 | 0.0074 | 50h | 1.0K |
| 5 | - | 0.27 | 0.0044 | 0.0042 | 0.0058 | 0.0077 | 122h | 2.2K |
| 6 | - | 0.13 | 0.0044 | 0.0044 | 0.0058 | 0.0075 | 56h | 1.1K |
| 7 | - | 0.12 | 0.0044 | 0.0044 | 0.0058 | 0.0076 | 50h | 1.0K |
| 8 | - | 0.096 | 0.0044 | 0.0045 | 0.0058 | 0.0072 | 44h | 800 |
| 9 | - | 0.048 | 0.0046 | 0.0047 | 0.0060 | 0.0075 | 28h | 400 |
| 10 | - | 0.024 | 0.0047 | 0.0047 | 0.0062 | 0.0077 | 25h | 200 |

*Table 3.* Ablation studies on 3D positional encoding.

| No. | RoPE | RFF | HOMO ↓ | LUMO ↓ | E1-CC2 ↓ | E2-CC2 ↓ |
|---|---|---|---|---|---|---|
| 1 | ✓ | ✓ | 0.0042 | 0.0040 | 0.0058 | 0.0074 |
| 2 | ✓ | ✗ | 0.0044 | 0.0043 | 0.0059 | 0.0073 |
| 3 | ✗ | ✗ | 0.0047 | 0.0048 | 0.0066 | 0.0078 |

the full set of grid cells), while $L = 3$ indicates merging until convergence. For random sampling, we test different sampling ratios, ensuring that the number of sampled cells matches that of merged cells at different levels for a direct comparison of the two approaches. The results are presented in Table 2, where we can draw the following conclusions:

1. (No. 1-4) As the number of merging levels increases, the model's efficiency improves significantly while the performance is relatively stable. In particular, the merging-until-convergence strategy achieves the best efficiency without compromising accuracy.

2. (No. 5-10) As the sampling ratio decreases, the the model's efficiency improves significantly. However, when the number of sampled cells is less than 1K, the performance begins to drop quickly. The best tradeoff between efficiency and accuracy happens when sampling 800-1K grid cells.

3. (No. 2-7) Overall, the adaptive merging strategy performs slightly better than the random sampling strategy, though the difference is not significant. However, the key advantage of adaptive merging is its ability to adaptively adjust the number of grid cells without the need to the tune the sampling ratio.

**3D positional encoding.** In this section, we evaluate the contributions of the two 3D positional encoding methods proposed in Sec.3.4. Specifically, we consider two settings: (1) using only RoPE (No. 2), and (2) removing both PE (No. 3). For the latter setting, positional information is incorporated by simply adding the linear projection of the 3D position $c_i$ to the input embeddings $x_i$. The results are presented in Table 3. We can clearly see that both 3D positional encodings play a crucial role in enhancing the model performance.

### 4.4. Further analysis

In this subsection, we perform a more in-depth analysis of SpaceFormer via a set of carefully designed experiments, as presented in Table 4. These experiments can offer deeper insights into the underlying source of SpaceFormer's performance gains and further elucidate our major contributions.

**Base settings.** We train a purely "atom-based" SpaceFormer without using any non-atom cells and refer to this setting as S1. Then, we add two settings S2 and S3 according to Table 2 No. 10 and No. 4, respectively (we use "S" instead of "No." to distinguish from previous tables).

**Regarding FLOPs.** We first examine whether the performance improvement stems solely from increased computational costs (FLOPs). To achieve this, we train a significantly larger SpaceFormer model by scaling the model width by 4x from the baseline setting S1, resulting in setting S4. The model width is adjusted to ensure that the training speed becomes slightly slower than that of setting S3. As shown in Table 4, although enlarging the model size leads to notable performance gains, the results of S4 still fall short of S3 on most downstream tasks. This indicates that the observed performance improvements are not merely a consequence of higher computational costs.

**Can atom-based models achieve similar performance by simply using virtual points?** We answer this question by conducting a set of experiments using the atom-based model Uni-Mol, while varying the number of virtual points as described in Sec. 3.1. This corresponds to settings S6-S10 in Table 4. We can see that while the introduction of virtual points can improve the model performance on several tasks, there is still a significant gap between UniMol and SpaceFormer, so our improvement should not be simply attributed by using virtual points. The next experiment will further highlight another difference: the use of MAE.

**MAE vs. denoising.** As described in Sec. 3.5, MAE corresponds to a stronger pretraining task than denoising. To see the difference empirically, we design a variant of MAE where only atom cells are masked, so the model will not need to predict whether a masked cell contains atom.

*Table 4.* Comparison between SpaceFormer and Uni-Mol adding VPs. See Sec. 4.4 for details.

| No. | Model | non-atom cells/VPs | HOMO↓ | LUMO↓ | E1-CC2↓ | E2-CC2↓ |
|---|---|---|---|---|---|---|
| 1 | SpaceFormer | 0 | 0.0051 | 0.0052 | 0.0065 | 0.0077 |
| 2 | SpaceFormer | 200 | 0.0047 | 0.0047 | 0.0062 | 0.0077 |
| 3 | SpaceFormer | 1000 | 0.0042 | 0.0040 | 0.0058 | 0.0074 |
| 4 | SpaceFormer-Large | 0 | 0.0046 | 0.0045 | 0.0060 | 0.0073 |
| 5 | SpaceFormer-denoising | 1000 | 0.0044 | 0.0041 | 0.0059 | 0.0075 |
| 6 | Uni-Mol | 0 | 0.0052 | 0.0060 | 0.0067 | 0.0080 |
| 7 | Uni-Mol | 10 | 0.0052 | 0.0058 | 0.0063 | 0.0078 |
| 8 | Uni-Mol | 50 | 0.0053 | 0.0056 | 0.0063 | 0.0079 |
| 9 | Uni-Mol | 100 | 0.0053 | 0.0057 | 0.0065 | 0.0080 |
| 10 | Uni-Mol | 200 | 0.0052 | 0.0058 | 0.0067 | 0.0080 |

This corresponds to setting S5 in Table 4. We can see that S3 outperforms S5 in all four downstream tasks, thus confirming the benefits of using MAE pretraining.

**Scalability with the number of cells.** Finally, we show that owing to our proposed 3D PE, the model enjoys much better scalability than UniMol with the increase of cells. As illustrated in Fig. 4, the pretraining cost of UniMol scales quadratically to the number of sampled cells, while the curve of SpaceFormer tends to have a linear shape. This verifies the efficiency of our proposed linear-complexity PE.

## 5. Conclusion

In this paper, we introduce SpaceFormer, a novel MPR framework that incorporates the 3D space beyond atomic positions to enhance molecular representation. To effectively and efficiently process 3D space, SpaceFormer is composed of 3 key components: grid-based space discretization, grid sampling/merging, and efficient 3D positional encoding. The resulting framework can be easily integrated with MAE training. Extensive experiments validate the effectiveness and efficiency of SpaceFormer across various tasks. In future research, a promising direction is to extend SpaceFormer to force field tasks and larger systems, such as proteins and complexes.

## Impact Statement

This paper presents work whose goal is to advance the field of Machine Learning. There are many potential societal consequences of our work, none which we feel must be specifically highlighted here.

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

## A. Experiment Details

The pretraining settings are detailed in Table 5, the downstream finetuning settings in Table 6, and the downstream tasks in Table 7.

*Table 5.* Pretraining Settings

| Hyper-parameters | Value |
| --- | --- |
| Peak learning rate | 1e-4 |
| LR scheduler | Linear |
| Warmup ratio | 0.01 |
| Total updates | 1M |
| Batch size | 128 |
| Residual dropout | 0.1 |
| Attention dropout | 0.1 |
| Embedding dropout | 0.1 |
| Encoder layers | 16 |
| Encoder attention heads | 8 |
| Encoder embedding dim | 512 |
| Encoder FFN dim | 2048 |
| MAE-Decoder layers | 4 |
| MAE-Decoder attention heads | 4 |
| MAE-Decoder embedding dim | 256 |
| MAE-Decoder FFN dim | 1024 |
| Adam $(\beta_1, \beta_2)$ | (0.9, 0.99) |
| Gradient clip | 1.0 |
| Mask ratio | 0.3 |
| Cell edge length $c^l$ | 0.49Å |
| $c_m$ for in-cell position discretization | 0.01Å |
| Merge level | 3 |

*Table 6.* Fine-tuning Settings

| Hyper-parameters | Value |
| --- | --- |
| Peak learning rate | [5e-5, 1e-4] |
| Batch size | [32, 64] |
| Epochs | 200 |
| Pooler dropout | [0.0, 0.1] |
| Warmup ratio | 0.06 |

## B. More Experiments

### B.1. Benchmark on QM9 dataset

We have expanded our evaluation to include the HOMO, LUMO, and GAP tasks on the QM9 dataset, which is standard in molecular property prediction. Our model demonstrates better performance compared with the baselines, as detailed in Table 8.

*Table 7.* Summary information of the downstream datasets

| Category | Task | Task type | Metrics | # Samples | Describe |
|---|---|---|---|---|---|
| Computational Properties | HOMO | Regression | MAE | 20,000 | The highest energy molecular orbital that is occupied by electrons |
| | LUMO | Regression | MAE | 20,000 | The lowest energy molecular orbital that is not occupied by electrons |
| | GAP | Regression | MAE | 20,000 | The energy difference between the HOMO and LUMO |
| | E1-CC2 | Regression | MAE | 21,722 | The energy of the first excited state computed using the CC2 method. |
| | E2-CC2 | Regression | MAE | 21,722 | The energy of the second excited state computed using the CC2 method. |
| | f1-CC2 | Regression | MAE | 21,722 | The free energy of the first excited state computed using the CC2 method. |
| | f2-CC2 | Regression | MAE | 21,722 | The free energy of the second excited state computed using the CC2 method. |
| | Dipmom | Regression | MAE | 8,678 | Dipole moment |
| | aIP | Regression | MAE | 8,678 | Adiabatic ionization potential |
| | D3_disp_corr | Regression | MAE | 8,678 | D3 dispersion corrections |
| Experimental Properties | HLM | Regression | MAE | 3,087 | Human liver microsome stability |
| | MME | Regression | MAE | 2,642 | MDRR1-MDCK efflux ratio |
| | Solu | Regression | MAE | 2,713 | Aqueous solubility |
| | BBBP | Classification | AUC | 1,965 | Blood-brain barrier penetration |
| | BACE | Classification | AUC | 1,513 | Binding results of human BACE-1 inhibitors |

*Table 8.* Benchmark on QM9 dataset

| Model | HOMO ↓ | LUMO ↓ | GAP ↓ |
|---|---|---|---|
| 3D Informax | 0.000952 | 0.000794 | 0.00155 |
| Uni-Mol | 0.000857 | 0.000763 | 0.00151 |
| SpaceFormer | 0.000594 | 0.000544 | 0.00106 |

