# OpenReview forum: "Beyond Atoms: Enhancing Molecular Pretrained Representations with 3D Space Modeling"
_ICML.cc/2025/Conference — ICML 2025 poster_

### Official Review · Reviewer_9MWX · 2025-03-03

**Overall Recommendation:** 3

**Summary:**

The paper introduces a novel framework called *SpaceFormer*, which models molecular as three-dimensional “images” by discretizing 3D space into grid units. This approach captures the spatial context of molecular more comprehensively than traditional methods that treat them as discrete collections of atoms. The authors demonstrate that *SpaceFormer* significantly outperforms existing 3D molecular pre-trained representation models across various tasks, especially in data-limited scenarios, by effectively leveraging additional spatial information. Their findings also reveal that randomly sampled virtual points can enhance performance, motivating further exploration of systematic 3D modeling in molecular representation.

## update after rebuttal
I thank the authors for their efforts in rebuttal. After reading all comments and rebuttals, I think the author has roughly solved the doubts about motivation, and this is crucial. So I'm willing to update the score 2->3

**Claims And Evidence:**

Several of the paper’s claims are incorrect or not well-supported.

**Essential References Not Discussed:**

The main text cites many literature related to quantum mechanics, but the authors do not understand the data generation of the MoleculeNet dataset, that is, they have a wrong understanding of the physical meaning of the data. Please refer to [1].



[1] Montavon G, Rupp M, Gobre V, et al. Machine learning of molecular electronic properties in chemical compound space[J]. New Journal of Physics, 2013, 15(9): 095003.

**Experimental Designs Or Analyses:**

- The authors used the processed dataset but did not provide dataset details. The fairness of the comparative experiment needs to be further verified.
- The performance of the molecular experimental properties dataset in Table 1 is not optimal, and the authors did not analyze it further.

**Methods And Evaluation Criteria:**

The description of the method is not sufficiently detailed, with some parts being confusing.

**Other Comments Or Suggestions:**

1. Functions or mathematical symbols are distinguished by other fonts, e.g. Embed_t (·)
2. Table 2 extends beyond the boundaries of the double columns.

**Other Strengths And Weaknesses:**

**Strengths**:
- SpaceFormer improves performance by making effective use of additional spatial information and utilizing randomly sampled virtual points.
- The authors perform a comprehensive evaluation against baseline models across a variety of datasets. The experiment section is solid.
- The method in this paper is innovative.

**Weaknesses**:
- The motivation of the paper is questionable. The authors suggest that existing molecular representations focus only on atomic coordinates and overlook electron density information, leading to the introduction of virtual points for electromagnetic field representation. However, the electromagnetic field is inherently linked to the atoms themselves, and the coordinates in the benchmark are the result of the optimization of the atomic structure and contain the atomic interaction information. Thus, the motivation is not well-founded.
- The writing requires improvement; the methodology is difficult to follow, and many details are not clear.
- The description of the dataset is not sufficiently detailed, with some parts being confusing. The authors should provide statistical details of the benchmark. At the same time, authors should compare models on existing benchmarks.

**Questions For Authors:**

- The atomic type is generally the corresponding atomic mass, for example, O is 16, how to quantify the virtual point type as NULL?
- After adding the virtual point, is the SE-(3) equivariance still valid? What are the specific dimensions of each encoding molecule on the transformer?
- Positional relative encoding is generally used to represent the relative interaction between atoms, what is the specific physical meaning of the relative position representation of VP and atoms in this paper?

**Relation To Broader Scientific Literature:**

Not involved.

**Theoretical Claims:**

I double-checked the full text, including the appendix, and the author did not provide a statement of the theory.

---

> ### Author Rebuttal · Authors · 2025-04-01
>
> Due to word limit, we will not quote the original text but refer to the title of the reviewer's feedback content.
>
> **1. Regarding Claims And Evidence**
>
> Thank you for your insightful feedback. In our paper, we have provided substantial experimental evidence to validate our claims. For instance, our results demonstrate that incorporating virtual points enhances model performance and techniques like grid sampling or merging improve efficiency without sacrificing effectiveness. Moreover,  various components, like position encoding, contribute significantly to the model's performance improvements.
>
> **2. Regarding Methods And Evaluation Criteria and Weaknesses 2**
>
> Thank you for your valuable feedback. We will revise the manuscript to enhance the readability, clarify our methodology  and add more details to the main text. We will also make our code open source to enhance understanding.
>
> **3. Regarding Experimental Designs Or Analyses  and Weaknesses 3**
>
> - Thank you for your feedback. In Section 4.1, we detailed the dataset's origin, significance, processing, and splitting methods, and included units and sample numbers in Table 1. Additional data size, description, task types, and evaluation metrics are found in Table 5 in the Appendix. We will release the data processing script with the code. If any part of this explanation remains unclear, we would appreciate further clarification from the reviewer. Regarding fairness in comparative experiments, we conducted additional tests on the QM9 dataset. The results, which show SpaceFomer's superior performance, are included in the first response to reviewer sLbT.
>
> - Thanks for your suggestion. We would like to clarify that  our results in experimental property-related tasks remain superior (with 4 out of 5 tasks ranked in the top two). We acknowledge that the enhancements are less pronounced than in computational property tasks. This may be due to the complexity and variability of the measurement environments for experimental properties, affected by factors like temperature and pH, which can impact measurement stability and accuracy.
>
> **4. Regarding Essential References Not Discussed**
>
> Thank you for your feedback. We apologize for any confusion and appreciate the chance to clarify. In our paper, we briefly mention the MoleculeNet dataset in the footnote of Section 4.1, focusing on challenges with pre-trained models, as discussed in previous work. We did not explore the data generation process.
>
> If specific statements cause misunderstandings, please let us know so we can address them thoroughly. We value your insights and welcome continued dialogue to improve our manuscript.
>
> **6. Regarding Reviewer's Weaknesses 1  and  Question 3**
>
> Thanks for your thoughtful feedback. We appreciate the opportunity to clarify our motivation. We acknowledge that atom coordinates encapsulate significant information, but relying solely on them necessitates extensive computation to derive electron densities or potential fields. Our approach introduces grid cells beyond atoms, offering a computationally efficient means to incorporate additional spatial information. Furthermore, many computational simulation methods in physics like electronic density distributions and potential fields are functions of the entire 3D space, not just atom positions. This underscores our motivation to explore modeling beyond atoms.  The interaction between atoms and virtual points with physical positions can be seen as a way for the model to efficiently learn some intermediate representations of the space beyond atoms, approximate spatial field effects through learnable interactions, thereby enhancing its understanding of the overall spatial information. We acknowledge this is an intuitive understanding and plan to pursue more theoretical validation in the future.
>
> **7. Regarding Reviewer's Other Comments Or Suggestions**
>
> Thank you for your suggestion. We use bold fonts for vectors and text fonts for functions. Specifically, in Section 3.2, we denote the embedding function as "\text{Embed}_t(\cdot)", following LaTeX conventions.  According to ICML guidelines,  tables can span two columns if within page margins. We've ensured compliance and appreciate your feedback, hoping you'll reconsider.
>
> **9. Regarding Question 1**
>
> Thank you for your question. In our model, each atom type is assigned a unique identifier mapped to an embedding, independent of atomic mass. For the virtual point type, a 'NULL' type is introduced with a distinct embedding to differentiate it from other atom types, without any mass-related encoding.
>
> **10. Regarding Question 2**
>
> Thanks for your question. Since we divide the 3D space where the molecules are located into grids and obtain the final grid cells/virtual points by merging, this method is deterministic and therefore does not affect the SE-(3) equivariance. Regarding your question on dimensions, we have detailed these specifications for each layer in Table 5, located in  Appendix A.

---

> > ### Comment · Reviewer_9MWX · 2025-04-02
> >
> > Thanks to the author for the answer that solved my doubts. I think the author's response and the overall quality of the paper are enough for me to improve my score.

---

> > > ### Author Response · Authors · 2025-04-02
> > >
> > > Thank you very much for your thoughtful review and increasing your score. We truly appreciate the time and effort you put into reading and evaluating our paper.
> > >
> > > Your comments were insightful and helped us better understand how to improve our work.
> > >
> > > Thanks again for your valuable feedback and support.

---

### Official Review · Reviewer_f27K · 2025-03-09

**Overall Recommendation:** 3

**Summary:**

This paper introduces SpaceFormer, a Transformer-based molecular pretrained representation (MPR) model that explicitly encodes the 3D space surrounding molecules, going beyond traditional atom-only representations. The method discretizes the 3D space into grid cells, applying adaptive grid merging for efficiency, and employs masked autoencoder (MAE) pretraining. Extensive experiments demonstrate significant performance improvements over baselines (e.g., Uni-Mol [Zhou, 2023]) across 15 molecular property prediction tasks (Table 1). Ablations confirm the effectiveness of each component (Tables 2, 3, 4).

**Claims And Evidence:**

The main claim—that modeling space beyond atoms enhances molecular representations—is strongly supported by experiments. Empirical results (e.g., ~20% error reduction on key quantum property tasks like HOMO/LUMO) convincingly validate this claim (Table 1). Ablation studies (Table 4) show improvements aren't just due to increased capacity or random noise points, but genuinely from modeling spatial context.

**Essential References Not Discussed:**

See the answer for "Relation To Broader Scientific Literature"

**Experimental Designs Or Analyses:**

Experiments and analyses are well-executed. Results averaged over multiple seeds and ablation studies thoroughly validate the method (Tables 2, 4). Minor weaknesses include no detailed runtime analysis and limited discussion on tasks where improvements were small (e.g., BBBP).

**Methods And Evaluation Criteria:**

Methods and evaluations are appropriate and rigorous. The chosen tasks (quantum and experimental properties) test generalization in limited-data scenarios using OOD splits, effectively evaluating pretraining value. Baselines (Uni-Mol, GEM, 3D-Infomax) are strong, ensuring a fair assessment.

**Other Comments Or Suggestions:**

- Minor typos (e.g., “concergence” → convergence in second column of page 7) should be corrected.
- Clarify average token count per molecule after merging and positional encoding details for reproducibility.

**Other Strengths And Weaknesses:**

Strengths: Highly original idea; consistent improvements across diverse tasks; careful experimental design and thorough ablation studies.
Weaknesses: Computationally intensive due to large input size (though mitigated by merging); does not explicitly enforce equivariance; unclear why method underperformed on BBBP (Table 1).

**Questions For Authors:**

1. Why does SpaceFormer slightly underperform on BBBP compared to simpler baselines? Could it indicate an overfitting or a limitation in your method?
2. Have you considered equivariant architectures as baselines? Would these be complementary to your grid-based approach?
3. Could you briefly explain why random virtual points improved performance initially (Figure 2)? What might the model be capturing from these points?

**Relation To Broader Scientific Literature:**

The paper clearly positions itself relative to prior 3D MPR models (Uni-Mol, 3D-Infomax) and makes a novel contribution by explicitly modeling empty space. However, it omits discussion of earlier 3D grid-based methods like AtomNet [Wallach, 2015] or equivariant models such as SchNet [Schütt, 2017], which could provide additional context.

**Theoretical Claims:**

No major theoretical issues. Their choice of grid resolution (0.49 Å) ensuring one atom per cell is logical. Positional encoding is reasonable and supported by experimental gains (Table 3). The theoretical arguments (e.g., empty space containing meaningful fields) are scientifically sound though qualitative.

---

> ### Author Rebuttal · Authors · 2025-04-01
>
> **1. Regarding "Minor weaknesses include no detailed runtime analysis and limited discussion on tasks where improvements were small (e.g., BBBP)." and "Why does SpaceFormer slightly underperform on BBBP compared to simpler baselines? Could it indicate an overfitting or a limitation in your method?"**
>
> Thank you for your insightful feedback. Regarding the runtime analysis on time cost, in Section 4.4, we provide a detailed analysis comparing SpaceFormer with Uni-Mol. The pretraining cost of Uni-Mol scales quadratically with the number of sampled cells, whereas SpaceFormer exhibits a more linear scaling pattern.
>
> For runtime analysis on smaller improvements in BBBP,  one reason can be  the complexity and variability of the measurement environments for experimental properties, which can be easily influenced by external factors, like temperature and blood-brain barrier biofilm status for BBBP. Such variability can affect the stability of measurement results and may not accurately.
> Additionally, as mentioned in the footnote of Section 4.1, the MoleculeNet dataset for the experimental task (including BBBP) has several limitations,  including invalid structures and inconsistent chemical representations, which affect its ability to differentiate molecular pretraining models. We will incorporate this analysis into the main text of our paper.
>
> **2. Regarding "it omits discussion of earlier 3D grid-based methods like AtomNet or equivariant models such as SchNet, which could provide additional context."**
>
> Thank you for your insightful feedback. We appreciate your suggestion to discuss earlier 3D grid-based methods. AtomNet and SchNet both use 3D convolutional neural networks, with SchNet focusing on atom positions. In contrast, SpaceFormer employs a transformer-based architecture to capture interactions across global grid cells, extending beyond just atoms positions. We will incorporate this discussion in our paper.
>
> **3. Regarding "Computationally intensive due to large input size (though mitigated by merging)"**
>
> Thank you for raising this point. Firstly, the input size is smaller than expected due to grid merging or sampling strategies, which reduce the number of grid cells effectively while preserving model performance, as detailed in Table 2. Secondly, SpaceFormer efficiently handles computational complexity using FlashAttention and two 3D relative positional encodings, allowing our model's time to scale nearly linearly with sequence length, as shown in Figure 4.
>
> **4. Regarding "does not explicitly enforce equivariance and have you considered equivariant architectures as baselines? Would these be complementary to your grid-based approach?"**
>
> Thank you for your insightful suggestion. We would like to clarify that both Uni-Mol and Mol-AE, which are included in our baselines, consider equivariance. Initially, SpaceFormer was developed with reference to Uni-Mol. However, we observed that explicitly enforcing equivariance, as Uni-Mol does, leads to increased computational complexity. For instance, Uni-Mol's pair-wise Gaussian distance  results in a complexity of $\mathcal{O}(n^2)$. Consequently, SpaceFormer encodes this information more efficiently through  3D distance PE using RFF to approximate Gaussian distance, and 3D directional PE using RoPE to capture direction, although strict equivariance is not guaranteed. Additionally,  SpaceFormer also employs random rotation data augmentation to enhance robustness. Figure 4 illustrates SpaceFormer's superior efficiency compared to Uni-Mol, while Table 1 shows SpaceFormer outperforming other equivariant baselines, highlighting our model's robustness.
>
> **5. Regarding "Clarify average token count per molecule after merging and positional encoding details for reproducibility" and "Minor typos should be corrected"**
>
> Thank you for your insightful feedback. In Table 2, we present the average token count per molecule (avg. cells) after the merging/sampling process. We recognize the importance of clarity and reproducibility, and we will include additional details about the positional encoding in the next version. Additionally, we plan to open source our code to facilitate reproducibility.  We will correct the typo  in the main text, as well as any other minor errors.
>
> **6. Regarding "Could you briefly explain why random virtual points improved performance initially (Figure 2)? What might the model be capturing from these points?"**
>
> Thank you for your insightful question. The inclusion of virtual points with physical positions  serves to provide additional spatial context, allowing the model to develop intermediate representations that enhance its understanding of the overall spatial information. In addition, during pre-training, the task of identifying atom locations in 3D grid cells becomes more challenging, which ultimately improves the model's performance in downstream tasks. This approach helps the model capture spatial and positional information more effectively.

---

### Official Review · Reviewer_sLbT · 2025-03-12

**Overall Recommendation:** 3

**Summary:**

This paper tackles molecular pretrained representation (MPR) by arguing that previous methods which focused solely on atom positions and types miss crucial physical context in the surrounding 3D space. Motivated by physical principles (e.g., the presence of electron densities and electromagnetic fields), the authors first demonstrate that simply adding randomly sampled virtual points can boost performance. Building on this observation, they propose SpaceFormer, a Transformer-based framework that discretizes the entire 3D space into grid cells, employs grid sampling and adaptive grid merging to manage computational cost, and integrates efficient 3D positional encoding (extending RoPE and using random Fourier features). The model is pre-trained via a Masked Autoencoder (MAE) strategy and is shown to outperform several baseline MPR models on diverse downstream tasks covering both computational and experimental molecular properties.

**Claims And Evidence:**

Yes

**Essential References Not Discussed:**

More grid-based approaches should be discussed:

[1] Kosmala, Arthur, et al. "Ewald-based long-range message passing for molecular graphs." International Conference on Machine Learning. PMLR, 2023.

[2] Wang, Yusong, et al. "Neural P $^ 3$ M: A Long-Range Interaction Modeling Enhancer for Geometric GNNs." The Thirty-eighth Annual Conference on Neural Information Processing Systems.

**Experimental Designs Or Analyses:**

Yes

**Methods And Evaluation Criteria:**

The experimental downstream tasks should also include widely used benchmarks such as QM9 and MD17, which are standard in molecular property prediction. Moreover, since grid/mesh-based methods are particularly adept at capturing long-range interactions [1, 2], the inclusion of the MD22 dataset would further validate the model's ability.

Since the grid-based methods are very time-consuming, what is the time consumption of Spaceformer and UniMol-AE (w/o grids)?

[1] Kosmala, Arthur, et al. "Ewald-based long-range message passing for molecular graphs." International Conference on Machine Learning. PMLR, 2023.

[2] Wang, Yusong, et al. "Neural P $^ 3$ M: A Long-Range Interaction Modeling Enhancer for Geometric GNNs." The Thirty-eighth Annual Conference on Neural Information Processing Systems.

**Other Comments Or Suggestions:**

NA

**Other Strengths And Weaknesses:**

NA

**Questions For Authors:**

NA

**Relation To Broader Scientific Literature:**

Introduce a grid-based molecule pretraining method

**Theoretical Claims:**

Yes

---

> ### Author Rebuttal · Authors · 2025-04-01
>
> **1. Regarding "The experimental downstream tasks should also include widely used benchmarks such as QM9 and MD17, which are standard in molecular property prediction. Moreover, since grid/mesh-based methods are particularly adept at capturing long-range interactions [1, 2], the inclusion of the MD22 dataset would further validate the model's ability."**
>
> Thank you for your insightful feedback and for highlighting the importance of these benchmarks. Following your comments, we have expanded our evaluation to include the HOMO, LUMO, and GAP tasks on the QM9 dataset, where our model demonstrates better performance, as detailed in the following table.
> |      Model    |      HOMO     |    LUMO       |      GAP      |
> |---------------|---------------|---------------|---------------|
> |  3D Infomax   |   0.000952    |    0.000794   |    0.00155    |
> |     Uni-Mol   |   0.000857    |    0.000763   |    0.00151    |
> |   SpaceFormer |   0.000594    |    0.000544   |    0.00106    |
>
> We also attempted to evaluate our approach on the stachyose molecule from the MD22 dataset, following your suggestion and the references provided. This would further validate our model’s capability in capturing long-range interactions. However, the MD22 dataset primarily focuses on the prediction of energy and force, while our current model framework is geared towards property prediction. Adapting our model for force prediction requires additional development time. We kindly ask for your understanding and additional time to complete these experiments. We are committed to providing the results in the next discussion phase.
>
> We greatly appreciate your constructive critique, which has strengthened our analysis. Thank you for your understanding and support.
>
> **2.  Regarding "Since the grid-based methods are very time-consuming, what is the time consumption of Spaceformer and UniMol-AE (w/o grids)?"**
>
> Thanks very much for your question. We would like to clarify that the grid-based methods employed in this paper are optimized to be not very time-consuming. We have implemented two key strategies to enhance efficiency.
>
> Firstly, by merging or sampling grid cells, we significantly reduce the number of grid cells without compromising the model's performance. This is detailed in Table 2, which presents the number of grid cells after merging or sampling alongside the corresponding model performance.
>
> Secondly, our model framework is designed to handle long sequences efficiently. We employ FlashAttention and two effective 3D relative positional encodings, ensuring that the computational time increases almost linearly with sequence length, as illustrated in Figure 4.
>
> Additionally, Mol-AE builds upon Uni-Mol with atom-based MAE pretraining, where only 15% of atoms are dropped for reconstruction during pretraining, resulting in negligible speed differences between Mol-AE and Uni-Mol. We compare the time consumption of Spaceformer and Uni-Mol in Figure 4, which demonstrates our model framework's superior efficiency for processing long sequences.
>
> **3. Regarding "More grid-based approaches should be discussed:
> [1] Ewald-based long-range message passing for molecular graphs." International Conference on Machine Learning.
> [2] Neural P3M: A Long-Range Interaction Modeling Enhancer for Geometric GNNs"**
>
> Thank you for your valuable feedback. We appreciate your insightful suggestions and will incorporate a discussion on these methods in our paper.
> This addition will help provide a more comprehensive analysis of long-range interaction modeling in molecular graphs. We sincerely appreciate your guidance in improving our work.

---

> > ### Comment · Reviewer_sLbT · 2025-04-03
> >
> > Thanks for the reply. I would keep my positive score.

---

> > > ### Author Response · Authors · 2025-04-09
> > >
> > > Sorry for the late reply regarding experiments on MD22 dataset.
> > > Our current framework primarily focuses on molecular property prediction rather than force field modeling. As such, adapting the model to the MD22 benchmark, which emphasizes force field energy prediction, required non-trivial modifications. Initially, we explored incorporating force prediction via the gradient of energy with respect to atomic coordinates, as is standard in many MD22 baselines. However, due to limitations in the current implementation of Flash Attention, particularly challenges with the backpropagation of  the gradient of the first-order derivative, this approach proved difficult to implement efficiently within the rebuttal period.
> > >
> > > We then shifted to a direct force regression approach, which avoids gradient computations and is adopted by several force-specific models. Using this method, we performed initial evaluations and report the energy prediction results on the Stachyose molecule. Due to time and resource constraints, we were not able to fully tune hyperparameters or expand to the full MD22 benchmark. Nonetheless, these results serve as an encouraging starting point, and we see significant potential for further improvements with more focused development.
> > >
> > >
> > > |      Model            |   Energy MAE  |   Interaction |
> > > |-----------------------|---------------|---------------|
> > > |  sGDML                |   4.0497      |      -        |
> > > |  Equiformer           |   0.1404      |  two-bodies   |
> > > |  MACE                 |   0.1244      |  four-bodies  |
> > > |  ViSNet               |   0.1283      | triplet and quadruplet interactions |
> > > | ViSNet(Neural P$^3$M) |   0.0856      | triplet and quadruplet interactions |
> > > |  Uni-Mol              |   0.323       |  two-bodies   |
> > > |  SpaceFormer          |   0.151       |  two-bodies   |
> > >
> > >
> > > We would also like to clarify that our model, Spaceformer, is not tailored specifically for force field tasks. Unlike ViSNet or MACE, which are designed to capture higher-order atomic interactions (e.g., three-body or four-body terms), our framework approximates two-body interactions through a efficient transformer-based architecture with a complexity lower than $\mathcal{O}(n^2)$. Despite this, Spaceformer demonstrates comparable performance to strong baselines such as Equiformer that explicitly model two-body interactions.
> > >
> > > We understand the reviewer’s interest in grid/mesh-based methods such as Neural P$^3$M, which are effective at capturing long-range interactions. While Neural P$^3$M builds upon ViSNet and benefits from a force-specific design, our approach is built on Uni-Mol and demonstrates that incorporating grid cells beyond atoms can enhance energy prediction, aligning with the motivation behind Neural P$^3$M.
> > >
> > > We sincerely appreciate the reviewer for highlighting this important research direction. We are committed to continuing this line of investigation, and in the final version of the paper, we will expand our discussion on MD22, the treatment of long-range interactions, and connections to Neural P$^3$M.
> > >
> > > We kindly ask the reviewers to consider the substantial effort we invested during the rebuttal period to adapt our framework and generate additional results, despite the limited time and computational resources available.

---

### Decision · Program_Chairs · 2025-05-01

**Decision:**

Accept (poster)

**Comment:**

The paper presents a new representation learning framework for molecules based on explicitly modeling the surrounding 3D space. All the reviewers appreciated the idea and the provided experiments. Although some reviewers initially raised concerns regarding computational efficiency and specific dataset details, the authors provided thorough rebuttals and additional experiments that addressed these issues. Overall, I recommend acceptance for the paper.